# Concurrent Circulation of Canine Distemper Virus (South America-4 Lineage) at the Wild–Domestic Canid Interface in Aburrá Valley, Colombia

**DOI:** 10.3390/v17050649

**Published:** 2025-04-29

**Authors:** Carolina Rios-Usuga, Melissa C. Ortiz-Pineda, Sergio Daniel Aguirre-Catolico, Víctor H. Quiroz, Julian Ruiz-Saenz

**Affiliations:** 1Grupo de Investigación en Ciencias Animales—GRICA, Facultad de Medicina Veterinaria y Zootecnia, Universidad Cooperativa de Colombia, Bucaramanga 680002, Colombia; 2Centro de Atención, Valoración y Rehabilitación del Área Metropolitana del Valle del Aburrá, Medellín 050001, Colombia; 3Independent Researcher, Medellín 050001, Colombia

**Keywords:** canine distemper virus, interspecies transmission, crab-eating fox, phylogeny, Morbillivirus

## Abstract

Canine distemper virus (CDV) is the causative agent of a widespread infectious disease affecting both domestic and wild carnivores. Owing to its ability to cross species barriers and its high fatality rate in unvaccinated animals, CDV poses a significant conservation threat to endangered wildlife worldwide. To date, two distinct CDV lineages have been reported in Colombia, with cases documented separately in domestic dogs and wild peri-urban carnivores. This study aimed to detect and characterize the concurrent circulation of CDV in naturally infected domestic dogs and crab-eating foxes (*Cerdocyon thous*) from the same area in Colombia. Through molecular and phylogenetic analyses, we identified the South America/North America-4 lineage infecting both populations simultaneously. Our findings revealed high genetic variability, multiple virus reintroductions, and a close relationship with CDV strains previously detected in the United States. These results confirm the simultaneous circulation of CDV in the domestic and wildlife interface and underscore the urgent need for an integrated approach to CDV prevention and control involving both domestic and wildlife health interventions.

## 1. Introduction

Canine distemper virus (CDV) is a negative-sense RNA virus classified within the *Paramyxoviridae* family and Morbillivirus genus and includes viruses of significant epidemiological importance to both humans and animals [1,2]. In carnivores, clinical disease is characterized by moderate to severe respiratory signs, gastrointestinal disease, immune suppression, and/or neurological disease [3,4]. Phylogenetic and evolutionary analyses have characterized CDV into at least 21 major genetic lineages with strong geographical distributions [5]: America-1 (including vaccine strains), America-2, America-3, South America/North America-4, America-5, Canada 1 and 2, Asia-1 to Asia 6, Europe Wildlife, Arctic, Africa-1, Africa-2, Europe-1/South America-1, South America-2 and 3, and Rockborn-like [6,7,8,9,10,11,12].

CDV has been widely recognized as a conservation threat to wildlife, especially endangered species, because of its high pathogenic potential and high mortality rates [13,14]. CDV is a highly adaptable multihost pathogen that can emerge and reemerge at the interface between wildlife and domestic animals [15,16]. Its presence in wildlife has been well documented, even without concurrent infections in domestic populations, largely due to the distinction between specialist and generalist strains [17].

CDV outbreaks in wild animals have been found to have occurred at different times than those in domestic dogs, indicating that viral transmission is not solely dependent on domestic dog populations [18]. This suggests that CDV may establish persistent cycles within wildlife communities, potentially facilitated by interactions among multiple species and ecological factors that sustain its circulation independently of domestic reservoirs. In fact, CDV persistence in a given ecosystem is likely due to the presence of a metareservoir that consists of multiple interconnected carnivore populations composed of multiple species [15,17]. In the Americas, wildlife has been proposed as a key factor in the intercontinental spread of the South America/North America-4 CDV lineage [14]. This hypothesis is supported by the detection of the virus in wild animals in both the United States and Colombia, as well as its presence in domestic dogs across Colombia, Peru, and Ecuador, suggesting a complex transmission dynamic between wild and domestic hosts [19,20].

At least two lineages of CDV have been reported in Colombia [8]. The South America-3 lineage was reported in clinically sick dogs in 2014, and its distribution has been restricted to Colombia [10]. The South America/North America-4 lineage was reported later and has been confirmed as the virus causing disease in domestic [9] and wild canids [20] in distant areas of the country. To highlight the interaction and simultaneous circulation of CDV at the wild–domestic interface, this study aims to diagnose and phylogenetically characterize the virus in domestic dogs and crab-eating foxes exhibiting clinical signs of infection within the same geographic region.

## 2. Materials and Methods

### 2.1. Clinical Samples

Thirty-four domestic dogs showing clinical signs compatible with CDV infection were recruited from five veterinary hospitals in the Aburrá Valley, located in the Department of Antioquia, Colombia. Additionally, 13 adult crab-eating foxes (*Cerdocyon thous*) were included; these animals were rescued using mammal nets by the veterinary medical personnel of the Wildlife Rescue Unit of the Regional Environmental Authority, which were rescued in different localities of the Aburrá Valley in response to citizen reports made after observing wild individuals showing clinical signs, mainly neurological. All the sampled animals originated from the Aburrá Valley, which is a natural river basin and one of the most populous valleys of Colombia in the Andean Region; these valleys are 60 km (37 mi) long at a level of 1300 m (4300 ft) above sea level. The sampling locations and distribution of cases within the Aburrá Valley are illustrated in Figure 1.

A thorough physical examination was conducted on each animal by the attending veterinarian at their respective hospital. Samples, including serum, nasal and conjunctival swabs, and urine, were collected from each subject. CDV infection was diagnosed via a rapid immunochromatographic test (CDV Ag Test Kit^®^, Bionote, Hwaseong-si, Republic of Korea). The test was conducted according to the manufacturer’s instructions. In brief, a conjunctival swab was placed into a sample buffer and mixed thoroughly. Using the pipette provided in the kit, five drops of the prepared sample were dispensed into the designated sample slot on the test cassette. The results were then observed and interpreted within 10–15 min.

### 2.2. RNA Extraction and cDNA Synthesis

Viral RNA was extracted from 140 µL of the serum by using a QIAamp Viral RNA Mini Kit (QIAGEN^®^, Hilden, Germany), following the manufacturer’s recommendations. The RNA quality and quantity were determined via a NanoDrop™ One (Thermo Scientific, Wilmington, DE, USA). The quantified RNA was preserved at −80 °C. A RevertAid™ First Strand cDNA Synthesis Kit (Thermo Scientific^®^, Glen Burnie, MD, USA) was used for cDNA synthesis. In brief, the mixture included 1 µL of dNTP mixture (10 mM), 1 µL (100 pmol/µL) of random hexamers, and 13 µL (1–3 µg) of total RNA. The mixture was heated for 5 min at 65 °C and then placed on ice. The RT mixture contained 4 µL of Buffer 5x Reverse Transcriptase and 1 µL of RevertAid™ Premium Enzyme Mix. Reverse transcription was completed for 10 min at 25 °C, 30 min at 50 °C, and 85 °C for 5 min.

### 2.3. PCR and Sequencing

All samples were analyzed via real-time PCR targeting the *Morbillivirus* phosphoprotein (P) gene. Amplification was performed with the Power Up™ SYBR Green Master Mix Kit (Thermo Scientific^®^, Waltham, MA, USA) according to the manufacturer’s instructions. Each reaction included 5 µL of cDNA, 25 µL of 2× Master Mix, 15 µL of nuclease-free water, and 2.5 µL (10 µM) of each primer (Forward: ATGTTTATGATCACAGCGGT, Reverse: ATTGGGTTGCACCACTTGTC [21]. PCR amplification was carried out in a QuantStudio 3 thermocycler (Thermo Scientific^®^) under the following conditions: initial denaturation at 95 °C for 4 min, followed by 35 cycles of 95 °C for 30 s, annealing at 50.8 °C for 30 s, and extension at 72 °C for 1 min. Water was used as a negative control, while cDNA derived from a commercial vaccine (Nobivac Puppy—MSD Animal Health, Summit, NJ, USA) served as a positive control.

Amplification was performed with the Power Up™ SYBR Green Master Mix Kit (Thermo Scientific^®^, Waltham, MA, USA) following the manufacturer’s instructions. Five microliters of cDNA was included in the reaction mixture, which contained 25 µL of Master Mix (2×), 15 µL of nuclease-free water, and 2.5 µL (10 µM) of the primers (for ATGTTTATGATCACAGCGGT and Rev- ATTGGGTTGCACCACTTGTC). The samples were amplified in a QuantStudio 3 thermocycler (Thermo Scientific^®^) following the following thermal conditions: denaturation at 95 °C for 4 min, 35 cycles of 95 °C for 30 s, 50.8 °C for 30 s, and 72 °C for 1 min. Water was employed as a negative control, and cDNA prepared from a commercial vaccine (Nobivac Puppy—MSD Animal Health, Summit, NJ, USA) was used as a positive control.

For positive samples with CT values less than 30, the Fsp coding region was amplified via Maxima Hot Start Green PCR Master Mix (Thermo Scientific). The amplification was performed with the F5/R5 primers (forward: TGTTACCCGCTCATGGAGAT, reverse: CCAAGTACTGGTGACTGGGTCT), which flank the F gene [22]. Each PCR included 6 µL of cDNA, 25 µL of 2× Master Mix, 15 µL of nuclease-free water, and 2 µL (10 µM) of each primer. The thermal cycling conditions consisted of initial denaturation at 95 °C for 4 min, followed by 35 cycles of 95 °C for 30 s, annealing at 50.8 °C for 30 s, and extension at 72 °C for 2 min, with a final extension step at 72 °C for 10 min.

For PCR product visualization, 5 µL of the amplicon was subjected to electrophoresis on a 1.5% agarose gel (AGAROSE I™, Amresco, Solon, OH, USA) and stained with EZ-VISION™ (Amresco, Solon, OH, USA). The bands were visualized under UV light using the GelDoc™ XR+ System (Bio-Rad, Hercules, CA, USA). Amplicon size was determined via comparison with the GeneRuler™ 100 bp Plus DNA Ladder (Thermo Scientific^®^). For Sanger sequencing, PCR amplicons were purified and sequenced by Macrogen Inc. (Seoul, Republic of Korea) using an ABI3711™ automated sequencer.

### 2.4. Phylogenetic Analysis

The Colombian sequences were edited and assembled via SeqMan, which is part of the DNAStar Lasergene™ V15.0 software package (Madison, WI, USA). To assess sequence similarity, nucleotide BLAST (basic local alignment search tool) was employed to compare Colombian CDV sequences (Accession numbers PV455503–PV455511) with those available in the GenBank database. Phylogenetic analyses were conducted in MEGA™ 7, which incorporates at least two representative sequences from each CDV lineage across different geographical regions. Phylogenetic relationships were inferred from the nucleotide alignment of Fsp sequences using the maximum likelihood methods, as implemented in MEGA™ 7. The optimal nucleotide substitution model, identified by MEGA™ 7, was HKY + G, which was subsequently applied in the maximum likelihood analysis. The America-1 lineage served as the outgroup for rooting the phylogenetic trees, and consensus trees were visualized in FigTree software version 1.4.

## 3. Results

All animals that tested positive with immunochromatography were confirmed to be infected with CDV through qPCR analysis. The gender distribution within the studied population showed no significant differences, with 40% being male and 60% being female. The majority of dogs exhibited respiratory symptoms, followed by neurological and gastrointestinal signs, with prevalence rates of 38.2%, 35.4%, and 26.4%, respectively. The main respiratory signs were nasal secretion and the presence of respiratory sounds; the main neurological signs were convulsions, gait disturbances, and involuntary movements; and the main gastrointestinal signs were vomiting and diarrhea. Regarding age, 68% of the dogs were younger than 12 months, and only 52.9% had received at least one CDV vaccination.

Among the foxes, all individuals presented with dehydration, cachexia, conjunctivitis, and fever (40–41 °C). The neurological manifestations included vestibule-cerebellar ataxia, mild cephalic tilt, the loss of extensor tone of the limbs on the affected side, involuntary movements, generalized weakness, and hyperexcitability (see Appendix A). Owing to the severity of their condition, poor prognosis, and low likelihood of successful reintroduction into the wild, humane euthanasia was performed with sodium pentobarbital to prevent further suffering.

For phylogenetic analysis, 405 bp of the Fsp coding region was evaluated and amplified from all the rt-PCR samples. Due to poor sequence quality, only eight domestic dog and one crab-eating fox CDV sequences were included in the analysis. Interestingly, although two different lineages have been previously reported in the Aburrá Valley, the South America/North America-4 lineage was confirmed as the infecting lineage in domestic dogs and the wild crab-eating fox (Figure 2), even though there were no reports of close contact with any of the CDV-infected domestic dogs with wildlife and/or foxes.

The analysis of the Fsp coding region revealed a high degree of similarity with Colombian strains belonging to the North/South America-4 lineage, with nucleotide identities ranging from 87.41% to 96.79%. The closest match was to previously reported Colombian *Cerdocyon thous* CDV sequences, followed by strains from the United States within the same lineage. Phylogenetic analysis, which was based on maximum likelihood inference of the nucleotide alignment (Figure 2), confirmed the characteristic phylogeographical distribution pattern of CDV. Notably, our findings demonstrate that the Aburrá Valley domestic Dog/*Cerdocyon thous* CDV interface clusters within the South America/North America-4 lineage. As previously reported, Colombian sequences group within the same clade as strains from Ecuador and the United States (North America-4 lineage). Additionally, the phylogenetic tree suggests at least four separate introductions of the South America/North America-4 lineage into the Aburrá Valley, as evidenced by its paraphyletic distribution.

## 4. Discussion

The findings of this study highlight the complex dynamics of CDV transmission at the interface between domestic dogs and wild fauna, particularly *Cerdocyon thous* (crab-eating foxes), in the Aburrá Valley. The clustering of viral sequences from both domestic dogs and wild foxes within the South America/North America-4 lineage suggests an active viral exchange between these populations. This interaction aligns with previous reports indicating that urban and peri-urban areas facilitate CDV spillover between domestic and wild animals, posing a significant risk to wildlife conservation and public health [15,23,24]. Canine distemper has become one of the most important and devastating diseases for wildlife worldwide [11], and although outbreaks of CDV have been recorded in domestic and wild populations from Colombia, this is the first molecular and phylogenetic confirmation of CDV concurrent circulation between these two animal populations in the studied area.

The observed paraphyletic distribution in the phylogenetic tree suggests that CDV was introduced into the Aburrá Valley multiple times, likely through both natural wildlife movement and anthropogenic factors. The identification of at least four independent introductions of the South America/North America-4 lineage into the region reinforces the idea that CDV is not maintained through a single continuous transmission chain but rather through recurrent spillover events. Similar patterns have been documented in other regions, where CDV strains from different geographic locations converge due to host migration or human-facilitated dispersal, such as the pet trade and the translocation of infected animals [5]. Although the intercontinental distribution of CDV through the Americas is well known [9], recent reports suggest that human travel could be modifying the natural distribution of disease, favoring transcontinental CDV, as South American CDV lineages have emerged in other countries, such as Gabon [25], India [26], Nigeria [27], and Namibia [28].

The close genetic relationship between *Cerdocyon thous* and domestic dog CDV sequences suggests that the CDV found in these foxes originated from local domestic dog transmission events, and that coupled with the high adaptability of this wild carnivore to urban and peri-urban environments [29], this species could act as a reservoir and/or intermediate host, keeping the virus out of direct domestic dog transmission cycles. Previous studies have shown that *Cerdocyon thous* populations are highly susceptible to CDV and often serve as bridge species, facilitating viral transmission between different ecological settings [20,30]. This role is particularly concerning in urbanized landscapes where human activities increase the frequency of encounters between domestic and wild carnivores, increasing cross-species transmission, as has been recently reported for CDV in the southern United States, where increased human development and increased precipitation data have shown increased CDV risk [31]. A comparable scenario was documented in northern Colombia, where interactions between wild grey foxes (*Urocyon cinereoargenteus*) and domestic dogs facilitated the persistence of rabies virus transmission; frequent contact between foxes, farmers, pets, and livestock created favorable conditions for sustained transmission between wildlife and domestic animals, with potential spillover into the human population [32,33].

The presence of CDV in both domestic and wild populations underscores the need for integrated disease control strategies. Effective vaccination programs targeting domestic dogs in high-risk areas could serve as critical measures to reduce viral spillover into wildlife populations. Additionally, conservation efforts should focus on monitoring and mitigating CDV outbreaks in wild carnivores, especially in regions where endangered species may be affected [15,34]. The implementation of molecular surveillance programs could provide real-time insights into viral evolution and the emergence of new transmission hotspots, enabling more proactive responses.

Previous clinical and phylogenetic analyses of Colombian CDV strains [9,20] have demonstrated that the South America/North America-4 lineage exhibits an enhanced neurovirulence in both domestic dogs and *Cerdocyon thous*, consistent with findings from the present study. Moreover, this lineage has been characterized by the Hemagglutinin 530D/549Y mutational signature [9,20], supporting the hypothesis that wildlife serves as a critical reservoir for CDV in Colombia. This reservoir role may facilitate viral spillover into unvaccinated domestic dog populations or those with incomplete or waning immunization coverage.

Serological tests for the identification of neutralizing antibodies have revealed significant differences in antibody titers between the CDV South America/North America-4 strain and the America-1 strains commonly used in vaccines [35]. This discrepancy suggests that the immune response induced by existing vaccines may be less effective against other newly identified strains, such as the South America/North America-4 strain actively circulating in the country. Consequently, *Cerdocyon thous* may play a key role in transmitting the South America/North America-4 lineage to domestic dogs, potentially facilitating vaccine breakthrough infections. This scenario is particularly concerning for unvaccinated dogs or those with waning immunity, as they may be more susceptible to infection despite prior vaccination [36,37]. Further studies are needed to evaluate the extent of vaccine escape and assess whether modifications to current vaccination protocols and vaccines are necessary to improve protection against emerging CDV variants [38,39]. Vaccination against CDV in domestic dogs has been established for decades; however, the use of such vaccines in wildlife has not been widely distributed due to multiple gaps with respect to the immune response in wild species and the diverse and adverse responses of classical attenuated vaccines in such animals [37]. Although different vaccines have been evaluated in other foxes (red foxes, grey foxes, fennec, among others) [15], no vaccine has been evaluated in the *Cerdocyon thous*, which belongs to a completely different genus within the Canidae Family [29].

The *Cerdocyon thous* (also known as the crab-eating fox) is a medium-sized nocturnal carnivore that weighs between 3 and 8 kg and belongs to the *Canidae* family. Its adaptability, generalist behavior, and ability to thrive in human-modified environments increase its likelihood of coming into frequent contact with both humans and domestic animals [40,41]. Although it is a least-concern species [42], its biological behavior and its ecological adaptability, which allows it to be in urban, peri-urban, and wild environments, can make it a risk bridge for the transmission of CDV and other viral diseases to susceptible species and even species at risk of conservation that are in the wild or in Wildlife Regional Ambiental Authority centers [43]. In fact, the transmission of CDV from domestic dogs to endangered species has become a high research priority because of the critical consequences that it has for endangered individuals and their ecosystems [44,45]. In fact, recently in the central Andean region of Colombia, CDV has been reported infecting an Andean bear (*Tremarctos ornatus*), which is the only bear native to South America and the largest land carnivore in that part of the world, classified as a Vulnerable species [46], highlighting the critical role of CDV for interspecies barrier jump and the possible role of mesocarnivores as potential reservoirs for other species [5,13,31].

Future research should focus on the ecological, behavioral, and molecular factors driving the interspecies spread of CDV; exploring interactions between *Cerdocyon thous* and domestic dogs in urban settings can help identify high-risk transmission hotspots. Habitat overlap, resource competition, and seasonal movement patterns may influence cross-species transmission frequency [15,17]. Whole-genome sequencing of circulating CDV strains is crucial for tracking viral evolution, identifying mutations linked to increased virulence or vaccine escape, and assessing host-specific adaptations. Furthermore, understanding environmental reservoirs and indirect transmission pathways, such as contaminated food, water sources, or fomites, could provide valuable insights into CDV control at the interspecies interface. Effective CDV control will require interdisciplinary collaboration among virologists, ecologists, veterinarians, and public health authorities to develop comprehensive surveillance and prevent future outbreaks in both domestic and wild carnivore populations.

## 5. Conclusions

Although CDV was first identified in domestic dogs nearly a century ago, its ongoing evolution and ability to infect multiple host species continue to pose a significant threat to both domestic and wild carnivores. This persistent challenge remains a major concern for veterinarians, virologists, and conservationists worldwide. Our findings confirm the presence of the South America/North America-4 lineage of CDV in domestic dogs and *Cerdocyon thous*, highlighting the critical need for further studies that analyze the wild–domestic interface in viral spread at the ecological level. Effective CDV control strategies must incorporate a comprehensive understanding of these transmission dynamics, emphasizing the need for interdisciplinary collaboration among researchers, wildlife experts, and public health authorities to mitigate the impact of the virus.

## Figures and Tables

**Figure 1 viruses-17-00649-f001:**
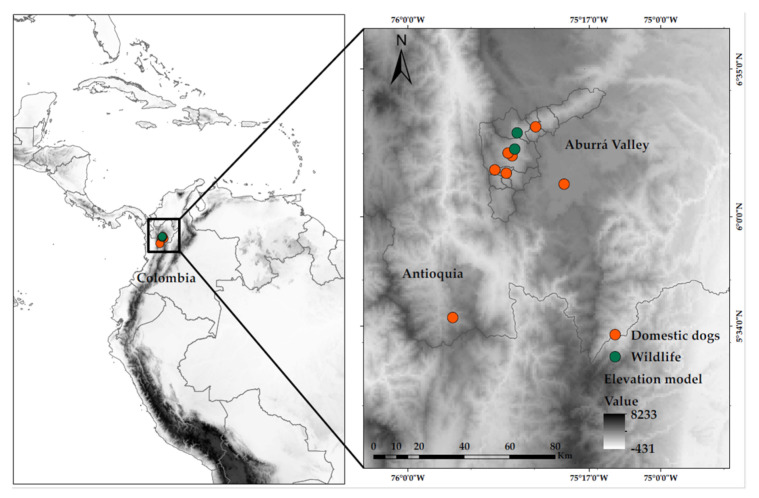
Map showing the location of CDV sampling centers in the Aburrá Valley, Antioquia, Colombia (red dots correspond to domestic veterinary services, and green dots correspond to the Regional Environmental Authority). The map was created in ArcGIS version 10.7.

**Figure 2 viruses-17-00649-f002:**
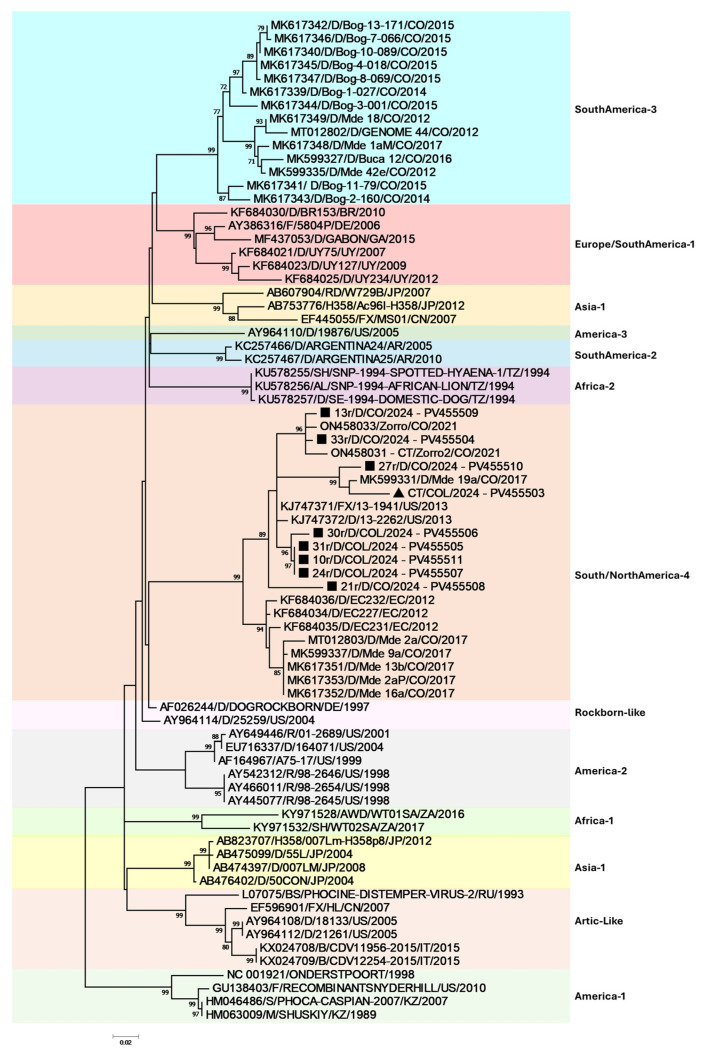
Phylogenetic maximum likelihood tree for the CDV Fsp region with 1000 bootstraps. The sequence labels include accession numbers, infected species, strains, countries, and years. Bootstrap support is indicated at nodes. The black squares denote the current Colombian dog sequences. The black triangle indicates the current *Cerdocyon thous* CDV sequence.

## Data Availability

All sequences are available at GenBank under accession numbers PV455503–PV455511.

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
