# Peer review of "Concurrent Circulation of Canine Distemper Virus (South America-4 Lineage) at the Wild–Domestic Canid Interface in Aburrá Valley, Colombia"

_viruses, 2025, doi:10.3390/v17050649_

Round 1
Reviewer 1 Report
Comments and Suggestions for Authors
General
This study discusses the concurrent circulation of CDV in infected domestic dogs and crab-eating foxes from the same area in Colombia. Unfortunately, there was only one sequence of a crab-eating fox available for a comparison with dog sequences. The conclusion that the clustering of viral sequences from both domestic and wild hosts within the South America/North America-4 lineage suggests active viral exchange between these populations (line 210-211) and CDV interconnections between these two animal populations in a small area (line 218-219) may therefore be a bit too strong. A few lines later (line 237) it is written “sustaining the virus outside of direct domestic dog transmission cycles”. The one sequence from a crab-eating fox is not evidence of a sustained spillover but the fact that many other crab-eating foxes tested positive is (unless these CDV positive foxes were infected with another lineage not occurring in dogs in this area).
Specific
Figure 1: The red and green dots are not the location of the animals but represent the location of the offices/clinics where the animals were submitted. The exact locations of the animals are not known?
Line 76: I can understand that dog owners will visit the veterinary clinics but how did they obtain free-roaming wild foxes? That is really interesting, where they captured by using nets and were they submitted at wildlife rehabilitation/refuge centers? May be the authors could add a few sentences on this?
Line 184: well, this lineage was found in one fox and not in foxes (plural)
Line 184-185: Transmission of rabies virus from domestic dogs to crab-eating foxes has been documented in Columbia. Hence, close contact between these two species seems to occur more often than thought.
Line 210-211: “The clustering of viral sequences from both domestic 210 and wild hosts” Why is ‘wild hosts’ plural, there was only one sequence from a crab-eating fox available?
Line 235-236: With only one sequence from a crab-eating fox available, this seems like ‘overrated’ conclusion. Based on the fact that another 12 foxes tested positive for CDV, it is a reasonable assumption but not supported by sequence data (genetic relationship).
Line 258-259: This sentence needs some more explanation. I may have overlooked it, but this is the first time the 530D/549Y mutation is mentioned (even without a reference) so the reader cannot do anything with this statement.
Line 263 – 265: not sure if this is written correctly. I assume that the authors mean that antibodies cross-neutralize not optimally against heterologous strains. Please make sure it is clear for the reader what is meant.
Line 279: ‘risk bridge’ – to which endangered species could the crab-eating fox transmit the virus, that would not be ‘reachable’ by domestic dogs, or do the authors mean that the virus could be transmitted from foxes to dogs and subsequently the dogs would infect other endangered species?
Author Response
General
This study discusses the concurrent circulation of CDV in infected domestic dogs and crab-eating foxes from the same area in Colombia. Unfortunately, there was only one sequence of a crab-eating fox available for a comparison with dog sequences. The conclusion that the clustering of viral sequences from both domestic and wild hosts within the South America/North America-4 lineage suggests active viral exchange between these populations (line 210-211) and CDV interconnections between these two animal populations in a small area (line 218-219) may therefore be a bit too strong.
R/: We agree with the reviewer. The sentence was rewritten to be clearer in the proposed idea.
A few lines later (line 237) it is written “sustaining the virus outside of direct domestic dog transmission cycles”. The one sequence from a crab-eating fox is not evidence of a sustained spillover but the fact that many other crab-eating foxes tested positive is (unless these CDV positive foxes were infected with another lineage not occurring in dogs in this area).
R/: We agree with the reviewer. The sentence was rewritten to be clearer in the proposed idea.
Specific
Figure 1: The red and green dots are not the location of the animals but represent the location of the offices/clinics where the animals were submitted. The exact locations of the animals are not known?
R/: Due to Personal Data protection Laws from Colombia, the exact location of the domestic dogs is not available; however, it is well known that sampled animals resided in areas close to or used live nearby to the veterinary clinics and hospitals (red dots in figure 1). And, as for the foxes (green dots), the marked dots refer to the location of the center where they were rescued and attended.
Line 76: I can understand that dog owners will visit the veterinary clinics but how did they obtain free-roaming wild foxes? That is really interesting, where they captured by using nets and were they submitted at wildlife rehabilitation/refuge centers? May be the authors could add a few sentences on this?
R/: We agree with the reviewer. Attached is more information about the rescue of the foxes rescued by the mobile wildlife unit of the local environmental authority
Line 184: well, this lineage was found in one fox and not in foxes (plural)
R/: We totally agree with the reviewer. wording is improved to provide clarity to the reader on the strain identified in the single sequence of the CDV-infected fox.
Line 184-185: Transmission of rabies virus from domestic dogs to crab-eating foxes has been documented in Columbia. Hence, close contact between these two species seems to occur more often than thought.
R/: We agree to the reviewer. A change was made in the wording of the text to clarify that the absence of contact with domestic dogs with wildlife refers exclusively to the dogs included in the study. Also, a short sentence about this topic was included in the discussion section.
Line 210-211: “The clustering of viral sequences from both domestic 210 and wild hosts” Why is ‘wild hosts’ plural, there was only one sequence from a crab-eating fox available?
R/: The wording of the lines (210-211) has been changed in order to clarify that this lineage was only found in several domestic dogs and in a wild fox.
Line 235-236: With only one sequence from a crab-eating fox available, this seems like ‘overrated’ conclusion. Based on the fact that another 12 foxes tested positive for CDV, it is a reasonable assumption but not supported by sequence data (genetic relationship).
R/: The wording of the lines suggested by the evaluator (235-236) is changed to strengthen the argument of the premise that is concluded regarding the genetic finding of the CDV sequences obtained.
Line 258-259: This sentence needs some more explanation. I may have overlooked it, but this is the first time the 530D/549Y mutation is mentioned (even without a reference) so the reader cannot do anything with this statement.
R/: We totally agree to the reviewer. The paragraph was rewrite for a better understanding of the idea.
Line 263 – 265: not sure if this is written correctly. I assume that the authors mean that antibodies cross-neutralize not optimally against heterologous strains. Please make sure it is clear for the reader what is meant.
R/: We agree to the reviewer. The wording of the lines suggested by the evaluator is strengthened in order to give clarity to the idea to be expressed associated with the difference in neutralizing antibodies of the strain used in vaccines vs. circulating strains.
Line 279: ‘risk bridge’ – to which endangered species could the crab-eating fox transmit the virus, that would not be ‘reachable’ by domestic dogs, or do the authors mean that the virus could be transmitted from foxes to dogs and subsequently the dogs would infect other endangered species?
R/: R/: We agree with the reviewer. The wording of the lines mentioned by the evaluator is strengthened, where the risk of CDV-infected foxes in wildlife and wildlife care centers is made clear.
Reviewer 2 Report
Comments and Suggestions for Authors
This manuscript outlines the concurrent circulation of Canine Distemper Virus (CDV) (South America-4 Lineage) at the Wild-Domestic Canid Interface in Aburrá Valley, Colombia. The study focusses on naturally infected domestic dogs and crab-eating foxes (Cerdocyon thous) from this region. The stated aim was to detect and characterize the concurrent circulation of CDV in naturally infected domestic dogs (34) and wild crab-eating foxes (13) using molecular and phylogenetic analyses. South America/North America-4 lineage was found in both domestic dogs and wild canid populations although it is stated that, ‘……….. Due to poor sequence quality, only eight domestic dog and one crab-eating fox CDV sequences were included in the analysis ………..’ . It is further suggested that the data presented indicates high genetic variability in isolates suggesting multiple virus re-introductions, and a close relationship with CDV strains previously detected in the United States but it isn’t clear that this can be concluded from the small number of samples analyzed, can this be explained further ?. Why was the study limited to crab-eating foxes ? Is there additional archival data available for other wild canid species in the region ? Is there any data from zoo canids or those in wildlife rehabilitation centers in other parts of Columbia ?
It is reasonably concluded that there is interspecies transmission at the domestic‒wildlife interface, especially with regard to the behaviour of the crab eating fox, and that the study indicates the need for an integrated approach to CDV prevention and control involving both domestic and wildlife health interventions. Additional discussion would add value to the manuscript. For example, the data presented does not in itself confirm close interaction between the domestic dogs and wild canids although it is likely a key route for CDV transmission in this study. Is there additional data available on CDV in other wild canids in the region ? Are vaccination campaigns planned in domestic dogs and wildlife species in this region of Columbia ? Which species of wild canids are of conservation concern ?
Author Response
This manuscript outlines the concurrent circulation of Canine Distemper Virus (CDV) (South America-4 Lineage) at the Wild-Domestic Canid Interface in Aburrá Valley, Colombia. The study focusses on naturally infected domestic dogs and crab-eating foxes (Cerdocyon thous) from this region. The stated aim was to detect and characterize the concurrent circulation of CDV in naturally infected domestic dogs (34) and wild crab-eating foxes (13) using molecular and phylogenetic analyses. South America/North America-4 lineage was found in both domestic dogs and wild canid populations although it is stated that, ‘……….. Due to poor sequence quality, only eight domestic dog and one crab-eating fox CDV sequences were included in the analysis ………..’ . It is further suggested that the data presented indicates high genetic variability in isolates suggesting multiple virus re-introductions, and a close relationship with CDV strains previously detected in the United States but it isn’t clear that this can be concluded from the small number of samples analyzed, can this be explained further ?. Why was the study limited to crab-eating foxes ? Is there additional archival data available for other wild canid species in the region ? Is there any data from zoo canids or those in wildlife rehabilitation centers in other parts of Columbia ?
R/: we agree with the reviewer that this information needs to be better presented. To date, there is only one scientific phylogenetic report of canine distemper virus (CDV) in wildlife in the country (also in cerdocyon thous), where the same strain identified in this study (South America/North America-4) was found. Also a case report paper has been published in a locally- indexed journal reporting the Presence of CDV in an Andean bear. (https://journal.espe.edu.ec/ojs/index.php/revista-serie-zoologica/issue/view/290)
Unpublished and personal reports show a high frequency wild animals arriving at rescue centers in the country that test positive for CDV , all either die or are euthanized due to their poor health condition. However, due to lack of publication of these data make it difficult to support. Therefore, our study also highlights the need to strengthen efforts to demonstrate the presence of CDV in other wild species, as well as to investigate its phylogenetics, clinical alterations, epidemiological risks, and impact on conservation.
We added some information related to this item in the discussion sector, including the reference to the Andean Bear infected with CDV
It is reasonably concluded that there is interspecies transmission at the domestic‒wildlife interface, especially with regard to the behaviour of the crab eating fox, and that the study indicates the need for an integrated approach to CDV prevention and control involving both domestic and wildlife health interventions. Additional discussion would add value to the manuscript. For example, the data presented does not in itself confirm close interaction between the domestic dogs and wild canids although it is likely a key route for CDV transmission in this study. Is there additional data available on CDV in other wild canids in the region ? Are vaccination campaigns planned in domestic dogs and wildlife species in this region of Columbia ? Which species of wild canids are of conservation concern ?
R/: we agree with the reviewer. As mentioned before there is a lack of published information regarding CDV in wildlife. However, some information regarding the presence of CDV in concern wildlife was added and discussed as much as the information regarding Vaccination against CDV.